# Characterisation of Phage Susceptibility Variation in *Salmonella*
*enterica* Serovar Typhimurium DT104 and DT104b

**DOI:** 10.3390/microorganisms9040865

**Published:** 2021-04-17

**Authors:** Manal Mohammed, Beata Orzechowska

**Affiliations:** School of Life Sciences, College of Liberal Arts and Sciences, University of Westminster, 115 New Cavendish Street, Fitzrovia, London W1W 6XH, UK; bea.orzech@gmail.com

**Keywords:** S. Typhimurium, DT104, DT104b, prophages, RMS, CRISPRs

## Abstract

The surge in mortality and morbidity rates caused by multidrug-resistant (MDR) bacteria prompted a renewal of interest in bacteriophages (phages) as clinical therapeutics and natural biocontrol agents. Nevertheless, bacteria and phages are continually under the pressure of the evolutionary phage–host arms race for survival, which is mediated by co-evolving resistance mechanisms. In Anderson phage typing scheme of *Salmonella* Typhimurium, the epidemiologically related definitive phage types, DT104 and DT104b, display significantly different phage susceptibility profiles. This study aimed to characterise phage resistance mechanisms and genomic differences that may be responsible for the divergent phage reaction patterns in *S*. Typhimurium DT104 and DT104b using whole genome sequencing (WGS). The analysis of intact prophages, restriction–modification systems (RMS), plasmids and clustered regularly interspaced short palindromic repeats (CRISPRs), as well as CRISPR-associated proteins, revealed no unique genetic determinants that might explain the variation in phage susceptibility among the two phage types. Moreover, analysis of genes coding for potential phage receptors revealed no differences among DT104 and DT104b strains. However, the findings propose the need for experimental assessment of phage-specific receptors on the bacterial cell surface and analysis of bacterial transcriptome using RNA sequencing which will explain the differences in bacterial susceptibility to phages. Using Anderson phage typing scheme of *Salmonella* Typhimurium for the study of bacteria-phage interaction will help improving our understanding of host–phage interactions which will ultimately lead to the development of phage-based technologies, enabling effective infection control.

## 1. Introduction

Non-typhoidal *Salmonella* (NTS) serovars predominantly cause a self-limiting diarrhoeal illness; however, they have recently observed to cause invasive extra-intestinal disease, including bacteraemia and focal systemic infections [1,2].

The global burden of disease caused by invasive NTS is significant and substantially exacerbated by the emergence of antibiotic resistant strains. The multidrug-resistant (MDR) *Salmonella enterica* serovar Typhimurium definitive phage type DT104 and the closely related DT104b are of considerable concern worldwide [3,4].

Interestingly, the emergence of MDR bacteria prompted a renewal of interest in bacteriophages (phages) as clinical therapeutics and natural biocontrol agents of foodborne pathogens, including *Salmonella* [5].

Although bacteria develop phage resistance mechanisms, phages continuously co-evolve to circumvent these anti-phage mechanisms [6,7,8]. Briefly, phage adsorption to a receptor (e.g., O-antigen lipopolysaccharide) on the cell surface is the initial step of the phage infection and host–phage interaction in *S.* Typhimurium [9]. Upon injecting its genetic material into the cytoplasm of the host cell, the phage follows either a lytic or lysogenic lifecycle. In the lytic cycle, the phage hijacks bacterial cell metabolic machinery to assemble virions and subsequently cause cell lysis. However, in the lysogenic cycle, the repressed phage genome integrates into the bacterial chromosome as a prophage, which provides the bacterium with a prophage-encoded phage resistance, superinfection exclusion (Sie), which prevents secondary infection with the same or closely related phage [8,10]. Nevertheless, the evolution of bacterial genomes allowed bacteria to acquire an array of mechanisms interfering with every step of phage infection. Blocking or modification of phage receptors, production of extracellular matrix, and production of competitive inhibitors enable bacteria to inhibit phage adsorption. However, when a phage succeeds in injecting its genome into a host cell, bacteria possess a variety of nucleic acid degrading systems that protect them from phage infection, such as restriction–modification systems (RMS) and clustered regularly interspaced short palindromic repeats (CRISPR), as well as CRISPR-associated (Cas) proteins [8,11,12,13].

Although MDR *S.* Typhimurium DT104 and DT104b are epidemiologically related, in the Anderson phage typing scheme [14], they display different phage susceptibility profiles, as shown in Table 1. We have previously shown that Anderson phage typing scheme of *S.* Typhimurium is a valuable model for study of phage host interaction [5] as whole genome sequencing (WGS) provided possible explanations for the difference in phage susceptibility among *Salmonella* Typhimurium DT8 and DT30. In this study, we performed comparative genome analysis on WGS data of representative strains of *S*. Typhimurium DT104 and DT104b to characterise phage resistance mechanisms and genomic differences that may impact the phenotypic lysis profiles. Understanding of host–phage interactions will ultimately lead to the development of phage-based technologies, enabling effective infection control. 

## 2. Materials and Methods

### 2.1. Bacterial Strains

Sequences of nine representative *S.* Typhimurium DT104 (*n* = 5) and DT104b (*n* = 4) were selected for the study. In addition to a well-studied *S.* Typhimurium strain, LT2 (DT4) was included in the comparative analysis. The whole genome sequences (WGSs) of all phage types were obtained from EnteroBase (https://enterobase.warwick.ac.uk/). The EnteroBase Barcodes and accession numbers of all strains as well as the year and country of isolation are presented in Table 2. All DT104b strains were isolated in Ireland in 2006, from swine (*n* = 2), the environment (*n* = 1), and food sources (*n* = 1). The DT104 strains were collected in the United Kingdom (*n* = 1), Ireland (*n* = 1), Germany (*n* = 1), and France (*n* = 1) over the 1975–2004 period. The DT104 strains were of human (*n* = 3) and bovine (*n* = 1) origin. The resistance profiles of all strains are provided in Appendix A. ResFinder 4.1 [15] was used to identify acquired antimicrobial resistance genes (ARGs) and chromosomal mutations, rendering *Salmonella* isolates, genotypically, resistant to antibiotics.

### 2.2. Identification of SNPs and Phylogenomics

The CSI Phylogeny 1.4 tool at the Center for Genomic Epidemiology [16] was used to identify single nucleotide polymorphisms (SNPs) and infer a phylogeny. The input parameters were as follows: (a) minimum depth at SNP positions at 10X; (b) minimum relative depth at SNP position at 10%; (c) minimum distance between SNPs (prune) at 10 bp; (d) minimum SNP quality score at 30; (e) minimum read mapping quality of 25; and (f) minimum Z-score (standard score) at 1.96. A phylogenetic tree was constructed based on the identified SNPs using FastTree. The Newick tree data were visualised on the MEGA X software [17]. The *Salmonella* Typhimurium strain LT2 was used to outgroup the phylogenetic tree.

### 2.3. Identification of Prophages, Plasmids, and R-M and CRISPR/Cas Systems

Lysogenic prophages integrated into *S*. Typhimurium genomes were determined using the web-based tool Phage Search Tool Enhanced Release, PHASTER [18], applying default parameters.

We constructed maximum likelihood (ML) phylogenetic trees based on the SNPs of detected prophages in all strains of DT104 and DT104b.

The identification and classification of the R-M systems were performed using Restriction-ModificationFinder 1.1, REBASE [19,20]. A threshold of 95% was selected for minimum percent identity (%ID) between the sequence in the input genome and the restriction enzyme gene sequence in Restriction-ModificationFinder 1.1; the selected minimum length was set at 60%. The Basic Local Alignment Search Tool (BLAST) on the National Center for Biotechnology (NCBI) website [21] was used to confirm prophages and the R-M systems.

Detection of CRISPR arrays and subtyping of Cas systems were performed on CRISPRFinder [22] and CRISPRCasFinder [23] web servers using default settings. In addition, the plasmid database; PLSDB [24] was used to identify plasmids present within the *S*. Typhimurium genomes. The analysis was completed using Mash (search strategy: mash screen) with a maximal *p*-value of 0.1 and minimal identity of 0.99. To remove redundancy from the output data, the winner-takes-all strategy was applied.

## 3. Results

### 3.1. Phylogenomics of S. Typhimurium DT104 and DT104b

Phylogenetic analysis was performed to predict genetic relatedness between *S*. Typhimurium strains. Figure 1 illustrates the phylogenetic neighbourhood, and Table 3 contains the SNP distance matrix among the *S.* Typhimurium genomes. Strains DP_F10, DP_N16, DP_N28, JE_2727, JM_04.26, MC_04-0529, R13 and DT104 reference strain (DT104 ref.) were found to be closely related; notably, DP_N16 and DP_N28 showed a close relation with a bootstrap value of 100% (Figure 1). In contrast, the DT104b reference strain (DT104b ref.) as well as the *S.* Typhimurium strain LT2 (DT4) displayed significant divergence among the studied genomes and were located on the same clade. Moreover, the SNP distance matrix exhibited an SNP divergence between TM75-404 and other studied genome sequences (Table 3).

### 3.2. Prophages in S. Typhimurium DT104 and DT104b

The PHASTER prophage analysis web server identified intact (score > 90), questionable (score 70–90), and incomplete phages (score < 70) in *S*. Typhimurium genomes [18]. Intact and questionable phages were confirmed by the BLAST database [21]; the confirmed prophages are shown in Table 4.

Phages Salmon_118970_sal3 (NC_031940) and Salmon_ST64B (NC_004313) were detected in all DT104b and DT104 strains but were absent in *S.* Typhimurium str. LT2 (DT4) which, unlike other strains, harbours Fels-1 (NC_010391) and Fels-2 (NC_010463). Gifsy-2 (NC_010393) and Gifsy-1 (NC_010392) were commonly shared between the studied genomes, whereas phage Entero_ST104 (NC_005841) was absent in the DT104b reference strain (DT104b ref.) and LT2 (DT4). In contrast, phage Salmon_SP_004 (NC_021774) was present in only DT104b ref. and phage Entero_lato (NC_001422) in the TM75-404 strain.

Phylogenetic analyses based on the SNPs of prophages of all strains of DT104 and DT10b showed that strains of DT104 are intermixed with the strains of DT104b (Appendix A).

Two different superinfection exclusion proteins (protein B and protein gp 17) were detected in all DT104 and DT104b strains, except one of the DT104b strains—DT104b ref (Appendix A).

### 3.3. R-M Systems S. Typhimurium DT104 and DT104b

As shown in Table 5, Restriction-ModificationFinder 1.1, REBASE [19,20] identified four R-M systems (Type I, Type II, Type III, and Type IV) in the studied genomes.

The Type I R-M system includes genes encoding restriction endonucleases, methyltransferases, and the specificity subunit (*S.StyUK1II)*. The presence and location of the putative Type I restriction enzyme *StyUK1IIP* were confirmed by BLAST [21]. The *StyUK1IIP* gene resides in the close proximity to the cognate methyltransferases (*M.SenTFII* and *M.Sen1899II)*; the percent identity was 99.97% (E value = 0.0) for the DT104b reference strain and 100% (E value = 0.0) for the remaining *S.* Typhimurium DT104b and DT104 strains. Putative Type I restriction enzyme *SenLT2IIP* was identified in the *S.* Typhimurium strain LT2 (DT4). The methyltransferase *M.SenTFII* was detected in all of the studied genomes except for the DT104b reference strain, which harbours *M.Sen1899II.* Nevertheless, both genes (*M.SenTFII* and *M.Sen1899II)* are closely related (99.94% percent identity, E value = 0.0) and code for the N6-adenine DNA methyltransferase subunit recognizing GAGNNNNNNRTAYG motifs.

The genes (*M.SenAboDcm, M.Sen641III, M.StyUK1V*) coding for methyltransferases, in the Type II R-M system, are common in all of the strains, except for LT2, which lacks methyltransferase *M.StyUK1V.* Moreover, Type IIG *Sty13348III* (restriction enzyme and methyltransferase) was identified in all of the genomes, excluding the LT2 and DT104b reference strains that instead harbour *StyUK1IV* Type IIG. The Type IIG *Sty13348III* and *StyUK1IV* genes are closely related (100% percent identity, E value = 0.0) and code for restriction endonuclease and methyltransferase recognising GATCAG motifs.

The genes of Type III and Type IV R-M systems were common for all studied *S.* Typhimurium genomes. The methyltransferase *M.StyUK1I* and restriction enzyme *SenAZII* were identified for Type III; for Type IV, only *StyLT2Mrr* was identified, which entails methyl-directed restriction.

### 3.4. CRISPR/Cas Systems in S. Typhimurium DT104 and DT104b

Two CRISPR loci (CRISPR-1 and CRISPR-2) and a type I-E Cas cluster were detected in all studied *S*. Typhimurium strains (Table 6). The type I-E CRISPR/Cas encompasses eight *cas* genes: *cas2*, *cas1*, *cas6*, *cas5*, *cas7*, *cse2*, *cse1* and *cas3* (Figure 2).

The CRISPR arrays in both loci share the same direct repeat sequence (29 base pairs). The CRISPR-1 arrays were identical in all DT104 and DT104b genomes, only with spacer variation in the DT104b reference strain, which harbours a unique spacer. Additionally, the DT104b reference strain and LT2 (DT4) share eight unique spacers positioned at the leader proximal end of the array (Figure 3). Although spacers within the CRISPR-2 loci of *S*. Typhimurium DT104b and DT104 showed high similarities, the length of arrays differed among the genomes with *n* = 22 and *n* = 12 spacers in MC_04-0529 and TM75-404, respectively, as well as *n* = 26 in DP_F10, DP_N16, DP_N28, JE_2727, DT104 reference, JM_04.26, and R13. Moreover, the CRISPR-2 loci of the DT104b reference contain *n* = 16 spacers, of which *n* = 7 are unique to DT104b ref. and LT2. The unique spacers were positioned internally (*n* = 6) and at the leader proximal end of the array (*n* = 1), as shown in Figure 4. The arrays within CRISPR-1 and CRISPR-2 were commonly shared between DP_F10, DP_N16, DP_N28, JE_2727, DT104 reference, JM_04.26, and R13.

### 3.5. Plasmids in S. Typhimurium DT104 and DT104b

PLSDB [24] identified ten plasmids among *S*. Typhimurium genomes, of which nine were detected in *S*. Typhimurium DT104 and DT104b (Table 7). The 93,939 base pairs (bp) virulence pSLT (NC_003277.2) plasmid was only detected in the *S.* Typhimurium strain LT2 (DT4).

Similar to DT104 and DT104b strains, LT2 carries the 33,784 bp pSE81-1705-3 (NZ_CP018654.1) plasmid as well as a megaplasmid (147,787 bp) that was first identified in the *Salmonella enterica* subsp. *Enterica* serovar Senftenberg strain NCTC10384, plasmid 3 (NZ_LN868945.1). The 94,045 bp pSC-09-1 (NZ_CP028319.1) plasmid was conserved in all DT104 and DT104b strains, except for the DT104b reference that harbours a linear 93,862 bp plasmid (NZ_LT855377.1). Significantly, the 3319 bp p1PCN033 (NZ_CP006633.1), carried by all DT104b strains (excluding DT104 or DT4), was absent in the DT104b reference strain.

#### Potential Phage Receptors

Analysis of potential genes coding for phage receptors revealed no differences among DT104 and DT104b strains (Appendix A).

### 3.6. WGS-Based Identification of Antimicrobial Resistance Determinants

The chromosomally located *AAC* (6’)-*Iaa* gene that renders amikacin and tobramycin ineffective was identified in all of the studied genomes. Seven *S.* Typhimurium strains exhibit an ACSSuT (ampicillin, chloramphenicol, streptomycin, sulphonamides, and tetracycline) penta-resistance profile, and these are DP_F10, DP_N16, DP_N28, JE_2727, MC_04-0529, R13, and the DT104 reference strain (Appendix A).

## 4. Discussion

Worldwide, infections and outbreaks caused by non-typhoidal *S.* Typhimurium represent a significant public health concern and an economic burden [25]. Zoonotic *S.* Typhimurium DT104, as well as DT104b, has a broad-host-range, and it is primarily transmitted through food sources and the faecal–oral route. The infection ranges from self-limiting gastroenteritis to invasive disease that requires prompt antibiotic therapy [1,2].

Nonetheless, selective pressure led to the emergence and subsequent dissemination of MDR *S.* Typhimurium DT104 and DT104b strains that phenotypically display an ACSSuT resistance profile. Moreover, recently, it has been reported that *Salmonella* has become increasingly resistant to other antibiotics of clinical significance, such as ciprofloxacin [26]. Limited treatment options for invasive and extraintestinal salmonellosis can lead to treatment failure and a surge in mortality and morbidity. The epidemic incidence and the global dissemination of MDR *Salmonella* prompted a renewal of interest in phages as clinical therapeutics and natural biocontrol agents [7,27,28]. Nevertheless, bacteria and phages are continually under the pressure of the evolutionary phage–host arms race for survival, which is mediated by the evolving anti-phage mechanisms in bacteria and parallel co-evolution of phage genomes [7,8,27].

Until recently, surveillance and outbreak investigation of *S.* Typhimurium was widely performed by phenotypic phage typing that relies on phage–host interaction. Advances in molecular biology and the emergence of WGS significantly improved the surveillance and outbreak investigation as well as the ability to comprehensively characterise the causative pathogen in silico [29,30,31].

The focus of this study was to characterise phage resistance mechanisms and genomic differences that may be responsible for the divergent phage reaction patterns in *S*. Typhimurium DT104 and DT104b.

SNPs were used to infer phylogenetic relationships between *S*. Typhimurium genomes as well as to assess the genetic diversity of the DT104 and DT104b strains. The phylogenetic analysis could not unambiguously differentiate phage types. Notably, the DT104b reference strain (DT104b ref) displayed significant divergence among the DT104b studied genomes. This may have been caused by the accumulation of SNPs in prophage regions or the *cas* genes but also the spacer variation of the CRISPR arrays and acquisition or loss of prophages and plasmids that result in the diversification of closely related bacteria [32,33,34,35].

Repressed and integrated into the bacterial chromosome, temperate phage genomes (prophages) provide the lysogen with a superinfection exclusion immunity. Through mechanisms such as blocking of phage DNA entry into the lysogen’s cytoplasm or inhibition of phage lysozyme, the superinfection immunity resists secondary infection by the same or closely related phages, providing the lysogen with a survival advantage [36]. Additionally, many phages carry virulence and antibiotic resistance genes [37].

The genomes of the strains analysed in this study possess multiple intact prophages that may have been inherited vertically or acquired due to the exposure to a multitude of bacterial viruses. Phages Salmon_118970_sal3 (NC_031940), Salmon_ST64B (NC_004313), Gifsy-1 (NC_010392), and Gifsy-2 (NC_010393) were highly prevalent among the *S.* Typhimurium DT104b and DT104 genomes; however, none of the strains harboured Fels-1 (NC_010391) and Fels-2 (NC_010463), which were detected in the *S.* Typhimurium reference strain LT2 (DT4). Nonetheless, the lysogen analysis did not reveal a significant difference in prophage profiles of DT104 and DT104b.

Moreover, SNP analysis of the detected prophages showed that strains of DT104 are intermixed with the strains of DT104b, indicating that prophages cannot explain the difference in bacterial susceptibility to typing phages.

Analysis of the genes coding for potential phage receptors revealed no differences in phage binding sites among DT104 and DT104b strains.

R-M systems allow bacteria to resist phage infection by degrading their DNA if recognised as foreign. They consist of methyltransferase and a cognate restriction endonuclease. The methyltransferase catalyses methylation of DNA to protect the self-genome from degradation by the restriction enzyme, which recognises and cleaves unmethylated (foreign) DNA at specific recognition sites. Based on the subunit composition and protein complexes, the R-M systems are classified into four types (I–IV) [38,39]. The in silico analysis and comparison of *S.* Typhimurium DT104 and DT104b R-M systems did not uncover possible explanatory causes of different lysis profiles. Although there were some variations in enzymes of Type I and Type II R-M systems of the DT104b reference strain, the detected methyltransferases and endonucleases were homologous to those of other DT104 and DT104b strains as well as LT2.

The CRISPR/Cas systems provide bacteria with adaptive and sequence-specific immunity against phages and plasmids. A CRISPR locus comprises a CRISPR array flanked by a *cas* operon. The CRISPR array is made up of short palindromic repeats (identical in length and sequence) that are interspaced by segments of DNA (spacers) derived from previous exposures to phages [8,40,41]. CRISPR/Cas loci are categorised into two distinct classes, class 1 CRISPR/Cas and class 2 CRISPR/Cas, as well as 5 types and 16 subtypes. The classification is based on the signature genes encoding interference modules and distinctive architecture of *cas* loci. The interference modules of the class 1 CRISPR/Cas systems contain multi-subunit crRNA–effector complexes, whereas class 2 systems possess a single subunit crRNA–effector module with a signature *cas9 gene* [40]. *S.* Typhimurium DT104 and DT104b harbour two CRISPR loci: CRISPR-1 and CRISPR-2. Located upstream of CRISPR-1 loci, eight signature *cas* genes (*cas2*, *cas1*, *cas6*, *cas5*, *cas7*, *cse2*, *cse1*, and *cas3*) belong to the class 1 CRISPR/Cas systems, type I subtype E (I-E). The type I-E system utilises multiprotein effector crRNA complexes, known as Cascade (CRISPR-associated complex for antiviral defence), to mediate interference of incoming nucleic acids. The Cascade complex consists of Cse1, Cse2, Cas7, Cas5, and Cas6 proteins, as well as crRNA. In type I systems, Cas6 processes pre-crRNA, resulting in intermediate crRNAs, and Cas 3 induces cleavage of the target DNA. Cas 1 and Cas 2 proteins, which are prevalent amongst the majority of CRISPR/Cas types, form a protein complex responsible for the incorporation of protospacers into the CRISPR array [40,41].

A comparison of CRISPR arrays of *S.* Typhimurium DT104 to CRISPR arrays of the DT104 reference strain, as well as a comparison of DT104b to CRISPR arrays of the DT104b reference strain, revealed potential loss and gain of spacers within CRISPR loci of DT104 and DT104b. Although the CRISPR-1 arrays were identical in all DT104 and DT104b genomes, the comparison of CRISPR-1 arrays of DT104b to the DT104b reference implies that DT104b lost nine contiguous internal spacers. In contrast, the composition of CRISPR-2 arrays within DT104b genomes suggests greater exposure to multiple phages compared to the DT104b reference strain. Interestingly, spacer duplication in CRISPR-2 was observed in the majority of studied genomes. The analysis of CRISPRs did not provide a possible explanation for the differing phage susceptibility of DT104 and DT104b. However, the composition of CRISPR arrays partially reflects the phylogenetic distances between the *S.* Typhimurium genomes, where strains possessing identical CRISPR loci were closely related. Nonetheless, Shariat et al. (2015) proposed that *Salmonella* CRISPR/Cas systems ceased its immunogenic function [42].

Acquired through horizontal gene transfer (HGT), plasmids (the elements of an accessory genome) provide bacteria with adaptive traits that can advantage bacteria under certain circumstances and stressors, such as antibiotics. The literature suggests that besides carrying virulence and antibiotic resistance genes, plasmids may also encode active R-M systems against phages [5,43]. Alternatively, the possession of conjugative antibiotic resistance plasmids may render bacteria susceptible to phages [44]. In this study, the distribution of plasmids amongst the DT104 and DT104b strains was assessed to predict whether these plasmid profiles could have impacted phage susceptibility patterns. Notably, unlike LT2 (DT4), DT104, and the DT104b reference, the DT104b strains harbour a low-molecular-weight (3319 bp) p1PCN033 plasmid, which has been associated with virulence and resistance traits [45]. However, it is unknown whether the p1PCN033 plasmid has a role in bacterial phage susceptibility or resistance; further studies are required to determine genetic markers that may be responsible for host–phage interaction.

## 5. Conclusions

In silico analysis of WGS of well-documented *S.* Typhimurium phage types DT104 and DT104b revealed no unique genetic determinants that might explain the variation in phage susceptibility among the two different phage types. However, this pilot study corroborates the complex dynamics of bacteria–phage interaction that limits conventional phage therapy. It also implies the necessity for further research, such as a study of host receptors involved in recognition and adsorption of phages, as well as phage counterstrategies to circumvent bacterial anti-phage mechanisms. Also, analysis of bacterial transcriptome using RNA sequencing will explain the differences in bacterial susceptibility and resistance to phages. Using Anderson phage typing scheme of *Salmonella* Typhimurium for the study of bacteria-phage interaction will help improving improving our understanding of host–phage interactions which will ultimately lead to the development of phage-based technologies, enabling effective infection control.

## Figures and Tables

**Figure 1 microorganisms-09-00865-f001:**
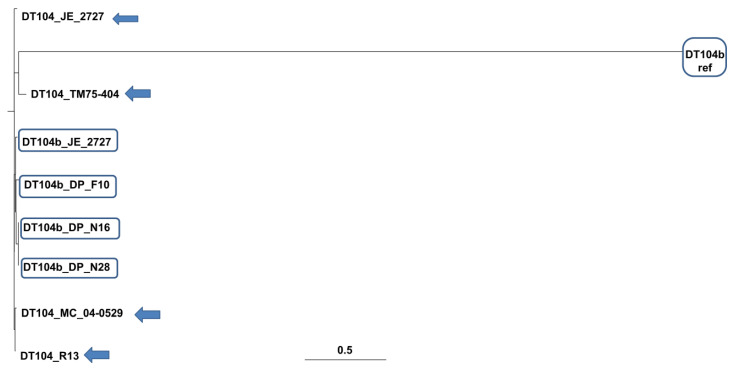
Maximum likelihood (ML) phylogenetic tree based on single nucleotide polymorphisms (SNPs) determined from the whole genome scheme 2. DT4 was used as a reference strain to construct the tree. The tree shows a close relation among DT104 strains (with arrows) and DT104b strains (boxed) with no significant divergence among them.

**Figure 2 microorganisms-09-00865-f002:**
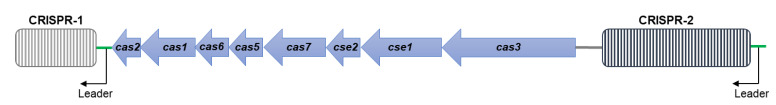
Schematic representation of the clustered regularly interspaced short palindromic repeats (CRISPRs) in *S*. Typhimurium DT4 (LT2), DT104, and DT104b. The eleven studied *S*. Typhimurium strains harbour two CRISPR loci, CRISPR-1 and CRISPR-2. The type I-E CRISPR/Cas system is encoded by an operon harbouring eight *cas* genes (blue boxed arrows) that are located upstream of CRISPR-1. Each CRISPR locus also contains a leader region (green horizontal lines). The black arrows show transcriptional orientation.

**Figure 3 microorganisms-09-00865-f003:**
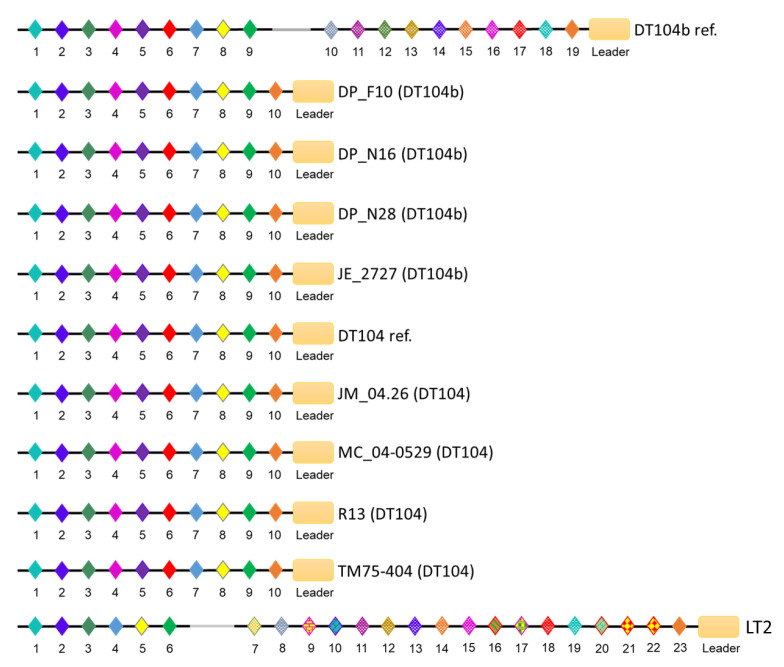
Schematic representation of CRISPR-1 locus in *S*. Typhimurium. The direction of spacers and repeats is shown, 5’ → 3’, with respect to the leader region (orange rectangle). The palindromic repeats (CGGTTTATCCCCGCTGGCGCGGGGAACAC) are shown as black horizontal lines. A distinctly coloured diamond represents each spacer. Spacer sequences and length were commonly shared between the DT104b and DT104 *S*. Typhimurium genomes, only with spacer variation in the DT104b reference (DT104b ref.) strain that possesses nine unique (in respect to all other strains) spacers represented as numbers 10–18. Additionally, *n* = 15 spacers in the DT104b reference genome and *n* = 7 spacers in the genomes of DP_F10, DP_N16, DP_N28, JE_2727, JM_04.26, MC_04-0529, R13, TM75-404, and DT104 ref. are identical to the spacers within the LT2 CRISPR-1 array.

**Figure 4 microorganisms-09-00865-f004:**
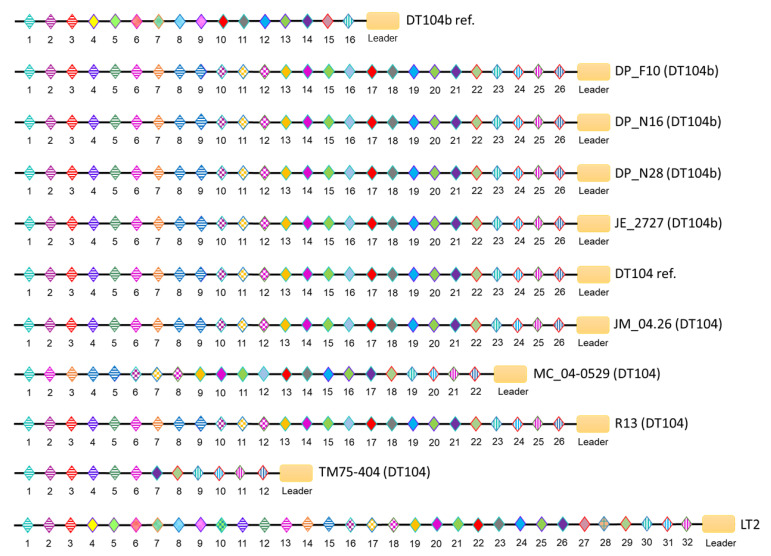
Schematic representation of CRISPR-2 locus in *S*. Typhimurium. The direction of spacers and repeats is shown, 5’ → 3’, with respect to the leader region (orange rectangle). The palindromic repeats (CGGTTTATCCCCGCTGGCGCGGGGAACAC) are shown as black horizontal lines. A distinctly coloured diamond represents each spacer. Strains DP_F10, DP_N16, DP_N28, JE_2727, DT104 reference (DT104 ref.), JM_04.26, and R13 harbour the same spacers (*n* = 26), of which *n* = 24 (spacers 1–8, 10–25; spacer 9 is a duplicate of spacer 8) are commonly shared with LT2 (DT4). The DT104 strains MC_04-0529 and TM75-404 possess shorter CRISPR-2 arrays, *n* = 22 and *n* = 12 spacers, respectively; *n* = 20 spacers (1–11, 21; spacer 5 is a duplicate of spacer 4) of MC_04-0529 and *n* = 11 spacers (1–11) of TM75-404 are identical to spacers within the LT2 CRISPR-2 locus. The DT104b reference (DT104b ref.) strain possesses *n* = 16 spacers, of which *n* = 7 (spacers 4–9, 15) were unique to DT104b ref. and LT2.

**Table 1 microorganisms-09-00865-t001:** Phage reaction pattern of the panel of 30 typing phages in *Salmonella* Typhimurium DT104 and DT104b. Variable degrees of reaction: −, no reaction; ±, weak reaction (1–20 plaques); +, 21–40 plaques; +++, 81–100 plaques; CL, confluent clear lysis; ±/−, variable reaction.

	1	2	3	4	5	6	7	8	9	10	11	12	13	14	15	16	17	18	19	20	21	22	23	24	25	26	27	28	29	30
DT104	-	-	-	-	-	-	-	-	-	-	CL	CL	-	-	-	-	CL	-	+	-	-	-	-	-	-	±/−	−/±	-	−/±	+/+++
DT104b	-	-	-	-	-	-	-	-	-	-	-	-	-	-	-	-	CL	-	±	-	-	-	-	-	-	±	±	-	-	+

**Table 2 microorganisms-09-00865-t002:** List of *S.* Typhimurium strains used in the study.

Strain ID	Phage Type	EnteroBase Barcode	Accession Number	Collection Year	Country of Detection	Source
DP_F10	DT104b	SAL_KA4333AA	ERS3655651	2006	Ireland	Swine
DP_N16	DT104b	SAL_KA4331AA	ERS3655649	2006	Ireland	Environment
DP_N28	DT104b	SAL_KA4341AA	-	2006	Ireland	Swine
JE_2727	DT104b	SAL_KA4322AA	-	2006	Ireland	Food
DT104b ref.	DT104b	-	-	-	-	-
JM_04.26	DT104	SAL_KA4067AA	-	2004	UK	Human
MC_04-0529	DT104	SAL_KA3878AA	-	2004	Ireland	Human
R13	DT104	SAL_KA3845AA	-	1999	Germany	Bovine
TM75-404	DT104	SAL_EA5197AA	-	1975	France	Human
DT104 ref.	DT104	SAL_EA9332AA	HF937208.1	-	-	-
LT2	DT4	-	AE006468.2	-	-	-

**Table 3 microorganisms-09-00865-t003:** SNP distance matrix among the *S.* Typhimurium genomes. The SNP differences between each pair of *S.* Typhimurium were determined by the CSI Phylogeny 1.4 tool [16].

	DP_F10	DP_N16	DP_N28	JE_2727	DT104b ref.	JM_04.26	MC_04-0529	R13	TM75-404	DT104 ref.	LT2
**DP_F10**	0	59	60	53	1095	72	76	59	139	88	655
**DP_N16**	59	0	5	50	1088	69	65	52	134	77	652
**DP_N28**	60	5	0	49	1087	70	66	53	133	78	651
**JE_2727**	53	50	49	0	1082	57	61	44	122	71	644
**DT104b ref.**	1095	1088	1087	1082	0	1075	1065	1064	1072	1059	862
**JM_04.26**	72	69	70	57	1075	0	58	39	117	66	639
**MC_04-0529**	76	65	66	61	1065	58	0	27	123	62	627
**R13**	59	52	53	44	1064	39	27	0	104	51	626
**TM75-404**	139	134	133	122	1072	117	123	104	0	105	636
**DT104 ref.**	88	77	78	71	1059	66	62	51	105	0	617
**LT2**	655	652	651	644	862	639	627	626	636	617	0

**Table 4 microorganisms-09-00865-t004:** List of confirmed prophages in *S.* Typhimurium strains. +, present; − absent.

Strain ID	Phage Type	Prophages, Accession Number								
		Salmon_118970_sal3 NC_031940	Salmon_ST64B NC_004313	Gifsy−2 NC_010393	Gifsy−1 NC_010392	Entero_ST104 NC_005841	Salmon_SP_004 NC_021774	Entero_lato NC_001422	Fels−1 NC_010391	Fels−2 NC_010463
DP_F10	DT104b	+	+	+	+	+	−	−	−	−
DP_N16	DT104b	+	+	+	+	+	−	−	−	−
DP_N28	DT104b	+	+	+	+	+	−	−	−	−
JE_2727	DT104b	+	+	+	+	+	−	−	−	−
DT104b ref.	DT104b	+	+	+	+	−	+	−	−	−
JM_04.26	DT104	+	+	+	+	+	−	−	−	−
MC_04−0529	DT104	+	+	+	+	+	−	−	−	−
R13	DT104	+	+	+	+	+	−	−	−	−
TM75−404	DT104	+	+	+	+	+	−	+	−	−
DT104 ref.	DT104	+	+	+	+	+	−	−	−	−
LT2	DT4	−	−	+	+	−	−	−	+	+

**Table 5 microorganisms-09-00865-t005:** Four distinct types of restriction–modification systems in the studied *S.* Typhimurium genomes. +, identified; − unidentified; *, undetermined.

Type I Restriction Modification System													
Gene	Function	Recognition Sequence	DP_F10 (DT104b)	DP_N16 (DT104b)	DP_N28 (DT104b)	JE_2727 (DT104b)	DT104b ref. (DT104b)	JM_04.26 (DT104)	MC_04-0529 (DT104)	R13 (DT104)	TM75-404 (DT104)	DT104 ref. (DT104)	LT2 (DT4)
*StyUK1IIP*	Restriction enzyme	GAGNNNNNNRTAYG	+	+	+	+	+	+	+	+	+	+	−
*SenLT2IIP*	Restriction enzyme	GAGNNNNNNRTAYG	−	−	−	−	−	−	−	−	−	−	+
*M.SenTFII*	Methyltransferase	GAGNNNNNNRTAYG	+	+	+	+	−	+	+	+	+	+	+
*M.Sen1899II*	Methyltransferase	GAGNNNNNNRTAYG	−	−	−	−	+	−	−	−	−	−	−
*S.StyUK1II*	Specificity subunit	GAGNNNNNNRTAYG	+	+	+	+	+	+	+	+	+	+	+
**Type II Restriction Modification System**													
*Sty13348III* Type IIG	Restriction enzyme/Methyltransferase	GATCAG	+	+	+	+	−	+	+	+	+	+	−
*M.SenAboDcm*	Methyltransferase	CCWGG	+	+	+	+	+	+	+	+	+	+	+
*M.Sen641III*	Methyltransferase	ATGCAT	+	+	+	+	+	+	+	+	+	+	+
*M.StyUK1V*	Methyltransferase	*	+	+	+	+	+	+	+	+	+	+	−
*StyUK1IV* Type IIG	Restriction enzyme/Methyltransferase	GATCAG	−	−	−	−	+	−	−	−	−	−	+
**Type III Restriction Modification System**													
*M.StyUK1I*	Methyltransferase	CAGAG	+	+	+	+	+	+	+	+	+	+	+
*SenAZII*	Restriction enzyme	CAGAG	+	+	+	+	+	+	+	+	+	+	+
**Type IV Restriction Modification System**													
*StyLT2Mrr*	Methyl-directed restriction enzyme	*	+	+	+	+	+	+	+	+	+	+	+

**Table 6 microorganisms-09-00865-t006:** CRISPR/Cas systems detected in the studied *S*. Typhimurium genomes.

Strain ID	Phage Type	Spacer Number			Cas Cluster Subtype
		CRISPR-1	CRISPR-2	Total	
DP_F10 (DT104b)	DT104b	10	26	36	I-E
DP_N16 (DT104b)	DT104b	10	26	36	I-E
DP_N28 (DT104b)	DT104b	10	26	36	I-E
JE_2727 (DT104b)	DT104b	10	26	36	I-E
DT104b ref. (DT104b)	DT104b	**19**	**16**	**35**	I-E
JM_04.26 (DT104)	DT104	10	26	36	I-E
MC_04-0529 (DT104)	DT104	**10**	**22**	**32**	I-E
R13 (DT104)	DT104	10	26	36	I-E
TM75-404 (DT104)	DT104	**10**	**12**	**22**	I-E
DT104 ref. (DT104)	DT104	**10**	**26**	**36**	I-E
LT2 (DT4)	DT4	23	32	55	I-E

**Table 7 microorganisms-09-00865-t007:** Distribution of plasmids identified among the studied *S.* Typhimurium genomes using the PLSDB webserver [24].

Plasmid	Accession Number	Length (bp)	DP_F10 (DT104b)	DP_N16 (DT104b)	DP_N28 (DT104b)	JE_2727 (DT104b)	DT104b ref. (DT104b)	JM_04.26 (DT104)	MC_04-0529 (DT104)	R13 (DT104)	TM75-404 (DT104)	DT104 ref. (DT104)	LT2 (DT4)
pSC-09-1	NZ_CP028319.1	94,045	+	+	+	+	−	+	+	+	+	+	−
p1PCN033	NZ_CP006633.1	3319	+	+	+	+	−	−	−	−	−	−	−
unnamed 2 *	NZ_CP043666.1	4593	+	+	+	−	−	−	−	−	−	−	−
pSE81-1705-3	NZ_CP018654.1	33,784	+	+	+	+	+	+	+	+	+	+	+
plasmid 3 *	NZ_LN868945.1	147,787	+	+	+	+	+	+	+	+	+	+	+
pCERC1	NC_019070.1	6790	−	+	+	−	−	−	−	−	−	−	−
Punnamed 4 *	NZ_CP036207.1	4149	−	−	−	+	−	−	−	−	−	−	−
plasmid: 2 **	NZ_LT855377.1	93,862	−	−	−	−	+	−	−	−	−	−	−
pAUSMDU00010534_03	NZ_CP045935.1	57,073	−	−	−	−	−	−	−	+	−	−	−
pSLT	NC_003277.2	93,939	−	−	−	−	−	−	−	−	−	−	+ ***

+, identified; − unidentified; unnamed 2 *, *Salmonella enterica* subsp. *enterica* serovar Kentucky strain 161,365 plasmid unnamed 2; plasmid 3 *, *Salmonella enterica* subsp. *enterica* serovar Senftenberg strain NCTC10384 plasmid 3; punnamed 4 *, *Escherichia coli* strain L725 plasmid unnamed 4; plasmid: 2 **, linear topology, *Salmonella enterica* subsp. *enterica* serovar Typhimurium isolate STMU2UK genome assembly, plasmid: 2 *** confirmed by PlasmidFinder-2.0 Server (https://cge.cbs.dtu.dk/services/PlasmidFinder/, accessed on 1 March 2021).

## Data Availability

The data that support the findings of this study are available from the corresponding author upon reasonable request.

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
