# Peer review of "Characterisation of Phage Susceptibility Variation in Salmonellaenterica Serovar Typhimurium DT104 and DT104b"

_microorganisms, 2021, doi:10.3390/microorganisms9040865_

Round 1
Reviewer 1 Report
General comment.
I read with interest the manuscript “Characterisation of phage susceptibility variation in Salmonella enterica serovar Typhimurium DT104 and DT104b” by Beata Orzechowska and Manal Mohammed. It is an interesting manuscript on comparative genomics of S. Typhimurium strains which aim is to understand better the genomic differences that may explain the phage-type differences among strains. This manuscript was written in a correct and clear English, with minimal typos and easy to follow and understand. I found the subject of the manuscript quite interesting since it is an attempt to understand better the strains biology and characterization taking into consideration a comparative approach, gene repertoire, and different biological elements that could provide their uniqueness. In the way that the manuscript has been written I realized the text is balanced more in favor of negative results than into make a detailed explanation of the differences they found. This lack of deeper analysis of the differences found and the more “negative” approach of the writing diminish the main message of the comparative approach. In addition, the lack of clear reference of what the starting point of this paper is precludes the assessment of the full contribution of this manuscript. Therefore, in my opinion, this manuscript needs a thorough revision on the way it has been written and, in some parts, needs a deeper analysis of the differences found between the studied strains. For all these reasons, I do not recommend this paper to publication as it is unless all these issues would be properly and thoroughly addressed.
Comments:
- In the abstract (P1, L21-24) the authors mentioned that this study’s aim is to characterize and find strain differences by “whole genome sequencing”. This text as it is in this part and after reading the text gives no clue if the authors themselves sequenced the strains for doing the analyses or if they used in house, public genomic data of this strains. If sequenced by themselves, it should be stated in materials and methods, otherwise it should be added in materials and methods that it is data obtained from a public database, giving the appropriate id’s of the source data they have used for these manuscript, for instance nucleotide data or nucleotide and functional annotation data. Without this information it is hard to evaluate the starting point of their analyses and the exact contribution of the authors in this comparative genomic manuscript.
- In Fig.1, the resulting tree should be outgrouped with the farthest strain in the analysis that should be LT2, however it seems that both LT2 and DT104b have been used to outgroup the tree. In any case, it should be stated in the text which is the branch used to outgroup the tree since this may change the topology of the tree.
- In the Results section, “Prophages in S. Typhimurium DT104 and DT104b”, the authors addressed the comparison based on the mobile elements on the strains. Unfortunately, they do not mention in the manuscript if the prophages in the different strains are located on the same genomic position on every strain compared. Sometimes this phages are hotspot of recombination in the hosts, it would be interesting to see if they are inserted in the same genomic positions in every case, and perhaps a more detailed analysis of the gene contents of the prophage differences between them. On the other hand, automatic programs to detect prophages may fail identifying small or remnant prophage sequences. The manuscript don’t mention if besides all automatic detection, there was a manual curation of the sequences to discard false negative detection of phages or remnants in the sequences studied. Same applies for the automatic detection of Restriction Modification and CRISPR-Cas System.
- In the same section of phages, there is no mention if the authors looked in detail for the prophage genes annotated as superinfection exclusion due to the importance of these genes to the trait the paper is focused on, that is the “phage-type” trait. No mention on the detailed analysis of differences between the different “sie” genes.
- Also for prophages there is no mention if analysis of possible lysogenic conversion genes could give a hint on the different phage types studied.
Specific changes
- P2, L36. Full stop not needed here.
- P2, L45. Typo in “anti-phage” (“ani-phage”)
- P2, L84. Coverage should be in capital “X” in “10x”
Author Response
Comments:
1.In the abstract (P1, L21-24) the authors mentioned that this study’s aim is to characterize and find strain differences by “whole genome sequencing”. This text as it is in this part and after reading the text gives no clue if the authors themselves sequenced the strains for doing the analyses or if they used in house, public genomic data of this strains. If sequenced by themselves, it should be stated in materials and methods, otherwise it should be added in materials and methods that it is data obtained from a public database, giving the appropriate id’s of the source data they have used for these manuscript, for instance nucleotide data or nucleotide and functional annotation data. Without this information it is hard to evaluate the starting point of their analyses and the exact contribution of the authors in this comparative genomic manuscript.
Thank you –We have clarified that sequencing data of well documented strains of DT104 and DT104b was obtained from public database.Kindly see all details within materials and methods section (Lines 72-74). Also, ID numbers and isolates details are provided in Table (2).
2.In Fig.1, the resulting tree should be outgrouped with the farthest strain in the analysis that should be LT2, however it seems that both LT2 and DT104b have been used to outgroup the tree. In any case, it should be stated in the text which is the branch used to outgroup the tree since this may change the topology of the tree.
Thank you –We have clarified this. Kindly see line (89-90). As illustrated in the tree, we found that one of DT104b strains (DT104b ref) is very divergent from other DT104b strains based on SNPs analysis (Fig. 1). Moreover, the tree shows that strains of DT104 are intermixed with DT104b strains.
3.In the Results section, “Prophages inS. Typhimurium DT104 and DT104b”, the authors addressed the comparison based on the mobile elements on the strains. Unfortunately, they do not mention in the manuscript if the prophages in the different strains are located on the same genomic position on every strain compared. Sometimes this phages are hotspot of recombination in the hosts, it would be interesting to see if they are inserted in the same genomic positions in every case, and perhaps a more detailed analysis of the gene contents of the prophage differences between them. On the other hand, automatic programs to detect prophages may fail identifying small or remnant prophage sequences. The manuscript don’t mention if besides all automatic detection, there was a manual curation of the sequences to discard false negative detection of phages or remnants in the sequences studied. Same applies for the automatic detection of Restriction Modification and CRISPR-Cas System.
Thank you –We have used a different database; PHAST; http://phast.wishartlab.com/index.htmlfor detection of prophages and found same prophages within bacterial genomes and all strains are lysogenic for same prophages detected here (kindly see table 4) –Comparative genomics revealed that prophages are inserted on the same position –Annotation of prophage genomes and analysis of their genes showed no differences in genes contents –SNP analysis of detected prophages showed that strains of DT104 are intermixed with the strains of DT104b indicating that propophages cannot explain the difference in bacterial susceptibility to typing phages (Supplementary file 2).
Kindly see added detailswithin methods (Lines: 94-95), results (Lines: 127-128), results (Lines: 227-229) and discussion (Lines: 282-284)
Analysis of CRISPRs, plasmids and RM systems among different strains of DT104 and DT104b showed no unique genetic determinants that can explain the difference in phage susceptibility among DT104 sand DT104b strains since most of the strains share identical CRISPRs, plasmids and RM systems... However, these findings highlights the complex dynamics of bacteria-phage interaction .. We are applying for funding to carry out further research to study host receptors involved in recognition and adsorption of phages in DT104 and DT104b strains.
4.In the same section of phages, there is no mention if the authors looked in detail for the prophage genes annotated as superinfection exclusion due to the importance of these genes to the trait the paper is focused on, that is the “phage-type” trait. No mention on the detailed analysis of differences between the different “sie” genes.
Thank you –We have detected superinfection exclusion protein (siB) within one prophage; Entero_ST104 which presents within all DT104 and DT104b except one strain of DT104b (DT104b ref.).
5.Also for prophages there is no mention if analysis of possible lysogenic conversion genes could give a hint on the different phage types studied.
Thank you –Comparative genomics revealed no significant difference among the prophages including lysogenic conversion genes –We agree that the difference in phage susceptibility among the two phage types (DT104 and DT104b) cannot be explained based on WGS data analysis –We are planning to carry out further experiments and transcriptomic analysis to characterise the basis of bacteria-phage interaction.
Specific changes
36.P2, L36. Full stop not needed here.Thank you –Done
37.P2, L45. Typo in “anti-phage” (“ani-phage”)Thank you –Done
38.P2, L84. Coverage should be in capital “X” in “10x”Thank you -Done
Reviewer 2 Report
Understanding of mechaniams of differences between bacterial strains to their susceptibility to phage infection is an important issue. Therefore, the topic of this paper is is scientifically sound. Using genomic analyses, the authors asked what are causes of different patterns of sensitivity to phages of two closely related Salmonella strains. Although the analyses are interesting, I found this paper incomplete. Even using the same type of methods, the conclusions could be significantly more supprted by results. Therefore, I recommend further analyses.
Specific comments:
- The authors found that analyses of restrictions systems, CRISPRs, prophages, and plasmid patterns do not allow to identify determinants responsible for susceptibility or resistance to different phages. However, genes coding for cell surface proteins which might serve as receptors to phages, as well as genes coding for proteins responsible for LPS synthesis (which can also serve as phage receptor), were not analyzed. I agree with the authors that assessing phage-specific receptors is necessary (lines 27-28 and 298-299) but this is what is lacking in this paper, although genomic data are available. Such analyses are mandatory to make this paper more complete.
- Line 52: replace "analogous" with "related"
- It is not clear whether genomes of analyzed phages were sequenced by the authors or they were taken from data bases. This must be indicated. Irrespective of the source of genomic data, accession numbers (along with data base name) of deposited and analyzed sequences must be provided.
- Table 1 is not cited in the text.
- In Discussion, it is not necessary to describe principles of restriction-modification systems (lines 227-238), the CRISPR/Cas system (lines 243-253) and basic information about plasmids (lines 281-287). These parts of Discussion can be deleted (perhaps they would be required in a BSc thesis, but not in a research article - all readers interested in this paper know what are restriction enzymes, plasmids and CRISPR/Cas).
Author Response
Specific comments:
1.The authors found that analyses of restrictions systems, CRISPRs, prophages, and plasmid patterns do not allow to identify determinants responsible for susceptibility or resistance to different phages. However, genes coding for cell surface proteins which might serve as receptors to phages, as well as genes coding for proteins responsible for LPS synthesis (which can also serve as phage receptor), were not analyzed. I agree with the authors that assessing phage-specific receptors is necessary (lines 27-28 and 298-299) but this is what is lacking in this paper, although genomic data are available. Such analyses are mandatory to make this paper more complete.
Thank you –we agree that assessing phage-specific receptors will provide insights into variations in phage susceptibility among DT104 and DT104b. And we are applying for external funding to carry out experimental analyses to investigate phage receptors. But, this paper is based on the analysis of the genomes of DT104 and DT104b to determine the possible role of certain genetic determinants involved in bacterial susceptibility to phages.
2.Line 52: replace "analogous" with "related"
Thank you –Done–kindly see line 52.
3.It is not clear whether genomes of analyzed phages were sequenced by the authors or they were taken from data bases. This must be indicated. Irrespective of the source of genomic data, accession numbers (along with data base name) of deposited and analyzed sequences must be provided.
Thank you –We have clarified that sequencing data of well documented strains of DT104 and DT104b was obtained from public database.Kindly see all details within materials and methods section (Lines 72-74). Also, ID numbers and isolates details are provided in Table (2).
4.Table 1 is not cited in the text.
Thank you –we have cited it (kindly see line 61).
5.In Discussion, it is not necessary to describe principles of restriction-modification systems (lines 227-238), the CRISPR/Cas system (lines 243-253) and basic information about plasmids (lines 281-287). These parts of Discussion can be deleted (perhaps they would be required in a BSc thesis, but not in a research article -all readers interested in this paper know what are restriction enzymes, plasmids and CRISPR/Cas).
Thank you –We have edited the discussion section as kindly suggestedand deleted theselines.
Round 2
Reviewer 1 Report
I would like to thank you the authors for the time and detail taken into the review of all my comments concerning the manuscript they submitted. Most of the comments have been correctly addressed and relevant changes have been added to the text of the manuscript. However, there are still two points that need to be further addressed:
Comment 2.
In Fig.1, the resulting tree should be outgrouped with the farthest strain in the analysis that should be LT2, however it seems that both LT2 and DT104b have been used to outgroup the tree...
The proper way to build a phylogenetic tree is adding one taxon that is phylogenetically distant from all the taxa involved in the study to cluster them. Due to the nature of the analysis done, all the taxa involved are phylogenetically close, so the outgroup (LT2) used is not distant enough to cluster the other taxa in the tree. In this case, the authors would need to include a different taxon to those used in this study to correctly outgroup the tree. In addition, although discussed in lines 210-213, the authors do not identify the reason(s) behind this unexpected result in the text. This answer can be directly obtained from the output of CSI Phylogeny 1.4 or contact the author of the software to discuss in detail the reason of this unexpected result.
Comment 3
In the Results section, “Prophages in S. Typhimurium DT104 and DT104b”, the authors addressed the comparison based on the mobile elements on the strains. Unfortunately, they do not mention in the manuscript if the prophages in the different strains are located on the same genomic position on every strain compared…
Automatic annotators are quite good to identify prophages, not perfect though. Sometimes remnants or incomplete phage content in a bacterial genome is not fully identified. To solve this, manual curation of the genome annotation helps to clarify the completeness of the prophage detection software used. For instance, the observation or detection of phage integrases and other phage related genes upstream and downstream these genes give an idea of possible “phage elements” missing in the automatic annotation, or confirm the proper negative phage content prediction.
Author Response
Comments:
1.In the abstract (P1, L21-24) the authors mentioned that this study’s aim is to characterize and find strain differences by “whole genome sequencing”. This text as it is in this part and after reading the text gives no clue if the authors themselves sequenced the strains for doing the analyses or if they used in house, public genomic data of this strains. If sequenced by themselves, it should be stated in materials and methods, otherwise it should be added in materials and methods that it is data obtained from a public database, giving the appropriate id’s of the source data they have used for these manuscript, for instance nucleotide data or nucleotide and functional annotation data. Without this information it is hard to evaluate the starting point of theiranalyses and the exact contribution of the authors in this comparative genomic manuscript.
Thank you –We have clarified that sequencing data of well documented strains of DT104 and DT104b was obtained from public database.Kindly see all details within materials and methods section (Lines 72-74). Also, ID numbers and isolates details are provided in Table (2).
2.In Fig.1, the resulting tree should be outgrouped with the farthest strain in the analysis that should be LT2, however it seems that both LT2 and DT104b have been used to outgroup the tree. In any case, it should be stated in the text which is the branch used to outgroup the tree since this may change the topology of the tree.
Thank you –We have clarified this. Kindly see line (89-90).As illustrated in the tree, we found that one of DT104b strains (DT104b ref) is very divergent from other DT104b strains based on SNPs analysis (Fig. 1). Moreover, the tree shows that strains of DT104 are intermixed with DT104b strains.
3.In the Results section, “Prophages inS. Typhimurium DT104 and DT104b”, the authors addressed the comparison based on the mobile elements on the strains. Unfortunately, they do not mention in the manuscript if the prophages in the different strains are located on the same genomic position on every strain compared. Sometimes this phages are hotspot of recombination in the hosts, it would be interesting to see if they are inserted in the same genomic positions in every case, and perhaps a more detailed analysis of the gene contents of the prophage differences between them. On the other hand, automatic programs to detect prophages may fail identifying small or remnant prophage sequences. The manuscript don’t mention if besides all automatic detection, there was a manual curation ofthe sequences to discard false negative detection of phages or remnants in the sequences studied. Same applies for the automatic detection of Restriction Modification and CRISPR-Cas System.
Thank you –We have used a different database; PHAST; http://phast.wishartlab.com/index.htmlfor detection of prophages and found same prophages within bacterial genomes and all strains are lysogenic for same prophages detected here (kindly see table 4) –Comparative genomics revealed that prophages are inserted on the same position –Annotation of prophage genomes and analysis of their genes showed no differences in genes contents –SNP analysis of detected prophages showed that strains of DT104 are intermixed with the strains of DT104b indicating that propophages cannot explain the difference in bacterial susceptibility to typing phages (Supplementary file 2).
Kindly see added detailswithin methods (Lines: 94-95), results (Lines: 127-128), results (Lines: 227-229) and discussion (Lines: 282-284)
Analysis of CRISPRs, plasmids and RM systems among different strains of DT104 and DT104b showed no unique genetic determinants that can explain the difference in phage susceptibility among DT104 sand DT104b strains since most of the strains share identical CRISPRs, plasmids and RM systems... However, these findings highlights the complex dynamics of bacteria-phage interaction .. We are applying for funding to carry out further research to study host receptors involved in recognition and adsorption of phages in DT104 and DT104b strains.
4.In the same section of phages, there is no mention if the authors looked in detail for the prophage genes annotated as superinfection exclusion due to the importance of these genes to the trait the paper is focused on, that is the “phage-type” trait. No mention on the detailed analysis of differences between the different “sie” genes.
Thank you –We have detected superinfection exclusion protein (siB) within one prophage; Entero_ST104 which presents within all DT104 and DT104b except one strain of DT104b (DT104b ref.).
5.Also for prophages there is no mention if analysis of possible lysogenic conversion genes could give a hint on the different phage types studied.
Thank you –Comparative genomics revealed no significant difference among the prophages including lysogenic conversion genes –We agree that the difference in phage susceptibility among the two phage types (DT104 and DT104b) cannot be explained based on WGS dataanalysis –We are planning to carry out further experiments and transcriptomic analysis to characterise the basis of bacteria-phage interaction.
Specific changes
36.P2, L36. Full stop not needed here.
Thank you –Done
37.P2, L45. Typo in “anti-phage” (“ani-phage”)
Thank you –Done
38.P2, L84. Coverage should be in capital “X” in “10x”
Thank you -Done
Reviewer 2 Report
The authors have addressed most of my previous comments, and I appreciate this. However, the authors declined to answer to point 1 of my previous review. Perhaps they missunderstood my intention. I did not ask for additional experimental work (which would be appropriate for a separate paper rather than for revision of this manuscript), but rather, I requested additional bioinformatic analyses, namely, the authors could analyse details of genes coding for potential phage receptors (surface proteins) or enzymes involved in synthesis of potential receptors (e.g. LPS). Since the authors already have appropriate data, this should not be a problem to preform such in silico analyses.
Author Response
The authors have addressed most of my previous comments, and I appreciate this. However, the authors declined to answer to point 1 of my previous review. Perhaps they missunderstood my intention. I did not ask for additional experimental work (which would be appropriate for a separate paper rather than for revision of this manuscript), but rather, I requested additional bioinformatic analyses, namely, the authors could analyse details of genes coding for potential phage receptors (surface proteins) or enzymes involved in synthesis of potential receptors (e.g. LPS). Since the authors already have appropriate data, this should not be a problem to preform such in silico analyses.
Thank you – as kindly suggested, we have analysed the genes coding for potential phage receptors and identified possible phage binding sites among all strains - Please see results within Supplementary table (2) in page 27 - We could not find differences in phage receptors among DT104 and DT104b strains — We have mentioned this within the abstract (lines 27 -28). Also, in the results section (lines 182-84) and in the discussion section (lines 237-238).